# Policies and strategies for HPV vaccination schedule completion in immunocompromised girls, including girls living with HIV: Qualitative insights from Eswatini, Malawi, and Uganda

Emily E. Crawford[1], Tosin F. Ajayi[1], Immaculate Ampeire[2], Michael Baganizi[2], Mike N. Chisema[3], Bridget C. Griffith[4], Lorraine Kabunga[5], Thuli Magagula[6], Lisa-Rufaro Marowa[7], Akachi E. Mbogu[1], Nobuhle Mthethwa[8], Stella Namutebi[5], Bhekiwe Shongwe[7], Timothy Tchereni[9], Frehiwot Birhanu[9☯], Xolisiwe Dlamini[6☯], Fredrick Luwaga[5☯], Sonali Patel[1☯*]

**1** Global Vaccines Delivery Team, Clinton Health Access Initiative, Boston, Massachusetts, United States of America, **2** Uganda National Expanded Programme on Immunization, Ministry of Health, Kampala, Uganda, **3** Expanded Program on Immunization, Ministry of Health, Lilongwe, Malawi, **4** Analytics and Implementation Research Team, Clinton Health Access Initiative, Boston, Massachusetts, United States of America, **5** Clinton Health Access Initiative, Kampala, Uganda, **6** Expanded Program on Immunization, Ministry of Health, Mbabane, Eswatini, **7** Clinton Health Access Initiative, Mbabane, Eswatini, **8** National AIDS Program, Ministry of Health, Mbabane, Eswatini, **9** Clinton Health Access Initiative, Lilongwe, Malawi

☯ These authors are joint senior authors.
* spatel@clintonhealthaccess.org

## Abstract

Immunocompromised girls, including girls living with HIV, face significantly increased risk for HPV-linked cervical cancer and require a differentiated human papillomavirus (HPV) vaccination schedule. While World Health Organization (WHO) vaccination guidelines exist, recommending immunocompromised individuals receive at least two, if possible three doses of HPV vaccine, little is known about how country health programs implement these recommendations. This study examines HPV vaccination policies, delivery strategies, barriers, and enablers for immunocompromised girls in Eswatini, Malawi, and Uganda—countries with a high burden of HIV and cervical cancer. A cross-sectional qualitative study was conducted through key informant interviews and focus group discussions with stakeholders from ministries of health, implementing partners, and health workers. A desk review of relevant policy documents was also conducted. Data were analyzed thematically to identify common and country-specific themes. All three countries follow the WHO's recommendation for a two-dose HPV vaccine schedule for immunocompromised girls. However, none have fully documented or consistently implemented policies or delivery strategies to reach immunocompromised girls with additional HPV vaccination doses. Stakeholder awareness of dosing schedules and strategies to vaccinate immunocompromised girls was limited and inconsistent, though higher in Eswatini than in Malawi or Uganda. Promising strategies were identified across the three countries, including use of Teen Clubs and adolescent HIV

**Data availability statement:** The qualitative data collected for this study includes sensitive personal information that could lead to identification of study participants. Data access requests can be made to the Clinton Health Access Initiative Scientific and Ethical Review Committee (research@clintonhealthaccess.org).

**Funding:** This study was funded by the Gates Foundation HAPPI Consortium (Grant INV-046461 INV-057603) via a sub-award to the Clinton Health Access Initiative through JSI Award Number 30142 (EC, TA, BG, LK, LR, AM, SN, BS, TT, FB, FL, SP). The manuscript's contents are the responsibility of the authors and do not necessarily represent the official views of the funders. The funders had no role in the study design, data collection and analysis, decision to publish, or preparation of the manuscript.

**Competing interests:** The authors have declared that no competing interests exist.

clinics to deliver HPV vaccines. Barriers included stigma, limited cold chain infrastructure, unclear operational policies, and weak data systems. Enablers included trusted health provider relationships, peer mentorship, and community awareness of cervical cancer risks. Improving HPV vaccine delivery for immunocompromised girls, including girls living with HIV, will require documenting and disseminating clear policies, integrating vaccination into HIV and adolescent health services, scaling successful strategies, and strengthening data systems to monitor coverage. Tailored, stigma-sensitive strategies are essential to ensure equitable cervical cancer prevention for immunocompromised adolescents in high HIV-burden settings.

## Introduction

Cervical cancer is a disease of global public health concern, with 661,021 new cases and an estimated 348,189 deaths recorded in 2022. It is the eighth most common cancer globally and the ninth leading cause of cancer deaths [1]. The disease is the leading cause of death by cancer in 37 countries and disproportionately affects women in low- and middle-income countries [1,2]. The primary cause of cervical cancer is the Human Papillomavirus (HPV) [3,4], which is also the most common sexually transmitted infection [5]. HPV is a primary cause of multiple types of cancer and is estimated to cause 4.5% of all cancers worldwide [6,7].

Women living with HIV (WLHIV) face an elevated risk of HPV infection and cervical cancer, compared to their HIV-negative peers [8–12]. The interaction of HPV and HIV have been described as a *syndemic*, or a synergistic epidemic, where the co-occurrence of the two viruses leads to worse outcomes, due to their interaction [13]. WLHIV face a six-fold higher risk of developing cervical cancer compared to those without HIV [13,14]. This is attributable to multiple factors, including elevated risk of infection with high-risk HPV serotypes, reduced clearance of HPV due to reduced immune response, increased risk of progression to cervical cancer, and a worse prognosis among WLHIV who develop cervical cancer [15,16]. WLHIV also face systematic barriers to cervical cancer prevention, screening, and treatment [17–19]. These risks extend to adolescent girls living with HIV (GLHIV), who face an elevated risk of HPV coupled with systematic barriers to HPV vaccination [20].

HPV vaccination is the main primary prevention intervention for cervical cancer and could prevent at least one third of HPV-related cancers [21,22]. In 2020, WHO adopted a Global Strategy to Accelerate the Elimination of Cervical Cancer as a Public Health Problem by 2030. This strategy incorporates the "90-70-90" targets, which aim for 90% of all girls to receive HPV vaccination by the age of 15 years, 70% of all women to undergo high-performance screening tests at least once by 35 years and again by 45 years, and 90% of all pre-cancers to be treated and invasive cancer cases to be managed by 2030 [22]. The strategy recommends HPV vaccine integration into all national immunization programs. To date, HPV vaccine implementation strategies in LMICs has primarily revolved around defining the target population, delivery methods, planning, communication, and coordination [23]. Strategies have rarely included discussion on vaccination strategies for GLHIV.

WHO currently recommends a two-dose vaccination schedule for immunocompromised girls, including GLHIV, with a third dose given where possible [21]. Immunocompromised individuals have a sub-optimal immune response to HPV vaccination [10,24,25] and they face barriers that lead to decreased HPV vaccine uptake [26,27]. Nevertheless, the HPV vaccine is considered safe and effective for these individuals, including GLHIV [10]. Multiple studies have investigated the appropriate dosing schedules as well as vaccine effectiveness for these groups [21,24,25]. Although ongoing clinical trials to evaluate the efficacy of a single-dose HPV vaccine in HIV-positive individuals may affect future vaccine recommendations [28,29], a differentiated dose schedule for GLHIV and other immunocompromised girls is currently considered essential. Despite this, there is almost no literature on how health systems can effectively deliver the recommended vaccination schedule to immunocompromised girls.

The syndemic of HPV, cervical cancer, and HIV disproportionately affects Southern and Eastern Africa [1]. Eswatini, Malawi, and Uganda are key countries for cervical cancer prevention; cervical cancer incidence per 100,000 women is 84.6 in Eswatini, 67.9 in Malawi, and 56.2 in Uganda [30] (Table 1). Cervical cancer mortality in these countries is also high, with the highest cervical cancer mortality rate in the world recorded in Eswatini [1,2,30]. Primary prevention of cervical cancer is particularly important in these countries, as access to cervical cancer prevention and treatment remains limited [32] while cervical cancer prevalence is increasing [33]. The need for adequate HPV vaccination is particularly acute for the estimated 30,000 GLHIV ages 10–14 in these countries, where HIV prevalence among this group is 7.5% in Eswatini, 2.4% in Malawi, and 2.2% in Uganda [31].

Nearly all countries in Southern and Eastern Africa offer routine HPV vaccination [34]; nevertheless, little is known about HPV vaccine coverage, vaccination strategies, or policies for immunocompromised girls; no peer-reviewed studies on these topics were identified in literature review. This paper seeks to describe the policies, strategies, barriers, and enablers for vaccinating immunocompromised girls, with a focus on GLHIV, in Eswatini, Malawi, and Uganda. The hypothesis for this research study is that, despite the availability of global recommendations, there are currently limited intentional efforts to reach HIV-positive and immunocompromised girls with full HPV vaccine schedules in these countries.

## Methods

### Ethics statement

This study was reviewed and approved by the Clinton Health Access Initiative (CHAI) internal Scientific and Ethical Review Committee (SERC). It was also approved by the Eswatini Health and Human Research Review Board (EHHRRB089/2024), the Malawi National Health Sciences Research Committee (NHSRC) (24/09/4522), the Uganda National Council for Science and Technology (UNCST) (HS4994ES), and the Makerere University School of Health Sciences Research and Ethics Committee (MakSHSREC) (MAKSHSREC-2024–744). Data collection was conducted under written informed consent.

The study took place in Eswatini, Malawi, and Uganda. In each country, data was collected through key informant interviews (KII), focus group discussions (FGD), and a desk review.

For the KII, purposive sampling was used to recruit stakeholders with direct experience in or knowledge of HPV vaccination efforts among adolescents living with HIV. In-country research team members supporting HPV vaccination

**Table 1. Cervical cancer and HIV burden in Eswatini, Malawi, and Uganda.**

|  | Eswatini | Malawi | Uganda |
|---|---|---|---|
| Age-standardized cervical cancer incidence per 100 000 women [30] | 84.6 | 67.9 | 56.2 |
| Cervical cancer mortality-to-incidence ratio [30] | 0.63 | 0.7 | 0.66 |
| Percent of women aged 30–49 years ever screened for cervical cancer [30] | 19 | 19 | 10 |
| Percent of girls ages 10–14 years living with HIV [31] | 7.5 | 2.4 | 2.2 |
| Estimated number of GLHIV ages 10–14 years [31] | 1,400 | 13,000 | 16,000 |

programs identified relevant stakeholders. At the national level, this included the Ministry of Health personnel, the Expanded Program on Immunization (EPI) manager, HIV program managers, and implementing partners supporting the HPV program. Sub-national stakeholders included immunization program managers, HIV program managers, and youth and adolescent program managers from urban and rural settings.

FGDs were conducted with health care workers in each country. Eligible participants included nurses and community health workers (CHW) from both urban and rural healthcare settings such as hospitals, clinics, ART centers, youth friendly clinics, and community health centers who have been working in health care for at least two years. Convenience sampling was used to identify the subnational geographies, specific clinics, and individual participants in the FGDs.

A desk review of documents relevant to each study country's policy on HPV vaccination for GLHIV and immunocompromised girls was conducted. The desk review included any official documents, including policy documents, official communications, strategy frameworks, etc., with information about vaccination schedules, HPV vaccination, GLHIV, or immunocompromised adolescents. Documents were identified by research team members with familiarity of the subject matter and were requested in each KII. All identified documents were read by a research team member, who extracted information relevant to the research questions as described above to a memo. Data from the desk review was used to confirm existence of written policies and strategies for vaccinating adolescent girls, GLHIV, and immunocompromised girls. This was used as a supplement to the qualitative data, which was used to understand the extent to which stakeholders were familiar with the policies and strategies for vaccination.

Data collection was conducted by trained CHAI personnel between November 2024 and January 2025. Data collection was conducted under informed consent using interview guides (for KII) and discussion guides (for FGDs), tailored to each country's context (S1 Text). The guides were developed using the Practical, Robust Implementation and Sustainability Model (PRISM) framework to understand contextual factors at multiple levels including the external environment, organizational characteristics and individual perspectives [35]. Guides were piloted in each country and adjusted for clarity. All KIIs and FGDs were recorded with the consent of the participants. Following the interview, the research assistants completed a detailed note-taking matrix, providing in-depth answers to each question discussed, and relying on the recordings to ensure completeness and correctness of the data. Due to the straightforward nature of the research questions, the capacity to capture detail with the structured notetaking forms, and time and human resource constraints, KIIs and FGDs were not transcribed [36,37]. All KIIs and FGDs were conducted in English.

Note-taking matrices were organized and analyzed following the principles of thematic qualitative analysis [38]. Note-taking matrices were compiled by country and type of data collection (KII and FGD) and coded using a hybrid inductive and deductive approach. A research team member (EC) identified overarching themes corresponding to the research question and PRISM factors [39,40]. To ensure data trustworthiness, the research team was composed of team members working directly on the research topic in each country (LK, LM, BS, TT, FB, FL), and authors engaged with data and participants through prolonged interaction and in-depth interviews. Member checking was conducted through sharing of initial results with stakeholder teams for validation. Due to limited time and human resources for analysis, manual analysis by the research team was complemented by exploratory thematic analysis carried out by ChatGPT (OpenAI, April 2024 version), a large language model. Data from the note-taking matrices was uploaded to ChatGPT, which was prompted to conduct a thematic analysis of the qualitative data. This analysis produced a list of themes. The research team member iterated these recommendations, reviewing the themes and codes, and verifying all analyzed data against the original note-taking matrices to ensure the validity of the suggested themes.

## Results

In total, 69 KIIs and 18 FGDs were conducted across Eswatini, Malawi, and Uganda between November 2024 and January 2025 (Table 2). KIIs in Eswatini were conducted with stakeholders from the Ministry of Health, other government entities, and implementing partners; three FGDs were conducted: one each with nurses, rural health motivators,

**Table 2. Data collection by respondent type and country.**

| | Eswatini | Malawi | Uganda |
|---|---|---|---|
| **Data Collection Period** | **November 2024** | **January 2025** | **December 2024** |
| *Key informant interviews* | | | |
| Government | 15 | 19 | 21 |
| Implementing Partners | 8 | 6 | 0 |
| **Total** | **23** | **25** | **21** |
| *Focus group discussions* | | | |
| Health workers | 2 | 3 | 12 |
| Community leaders | 1 | 0 | 0 |
| **Total** | **3** | **3** | **12** |

and community leaders. In Malawi, KIIs were conducted with stakeholders from the Ministry of Health, other government entities, and implementing partners, at national and sub-national levels (Lilongwe and Ntchisi districts) and FGDs were conducted with health care providers in three districts of Mzimba North, Mchinji and Machinga, drawn from a total of 23 health clinics, representing all the three administrative regions of the country. In Uganda, five KIIs were conducted at the national level, and 16 in Bududa, Kampala, Lira, and Rukungiri districts. Twelve FGDs were conducted with health care providers in clinics located in the same four districts.

## HPV vaccine introduction

KII respondents described the timeline for HPV vaccine introduction and their knowledge of vaccine dosage; member checking and the desk review (Table 3) confirmed timelines. Uganda introduced the HPV vaccine in 2015 with a two-dose schedule, as recommended by WHO at that time, and a third dose for immunocompromised girls (Fig 1). Malawi introduced the HPV vaccine with a two-dose schedule in a phased approach starting in 2018, with national coverage achieved in 2021. Eswatini introduced the HPV vaccine in 2023 with a single-dose schedule; a second dose for immunocompromised girls was introduced in 2024. Malawi shifted from a two-dose schedule to a single-dose schedule in 2024, with a second dose for immunocompromised girls. In 2025, Uganda shifted to a single-dose schedule with a second dose for

**Table 3. Summary of documents identified for the desk review.**

| Title | Author | Year | Document type |
|---|---|---|---|
| *Eswatini* | | | |
| Eswatini Vaccine Integration Framework V.01 2024–2026 | Eswatini Ministry of Health Expanded Programme on Immunization (EPI) | 2024 | Policy |
| *Malawi* | | | |
| EPI Comprehensive Multi-Year Plan 2016–2020 | Government of Malawi | 2015 | Strategy |
| *Uganda* | | | |
| Consolidated Guidelines for prevention and treatment of HIV and AIDS in Uganda | Ministry of Health | 2024 | Guidelines |
| Switching from HPV two dose to a one dose schedule and scale up cervical cancer screening | Director General Health Services, Ministry of Health | 2024 | Administrative communication |
| National Child Policy | Government of Uganda Ministry of Gender, Labour, and Social Development | 2020 | Policy |
| Post Introduction Evaluation of HPV, bOPV, IPV, Fridge-Tag in Uganda | Ministry of Health | 2017 | Evaluation Report |
| Uganda School Health Policy | Ministry of Health, Ministry of Education & Sports | 2008 | Policy |

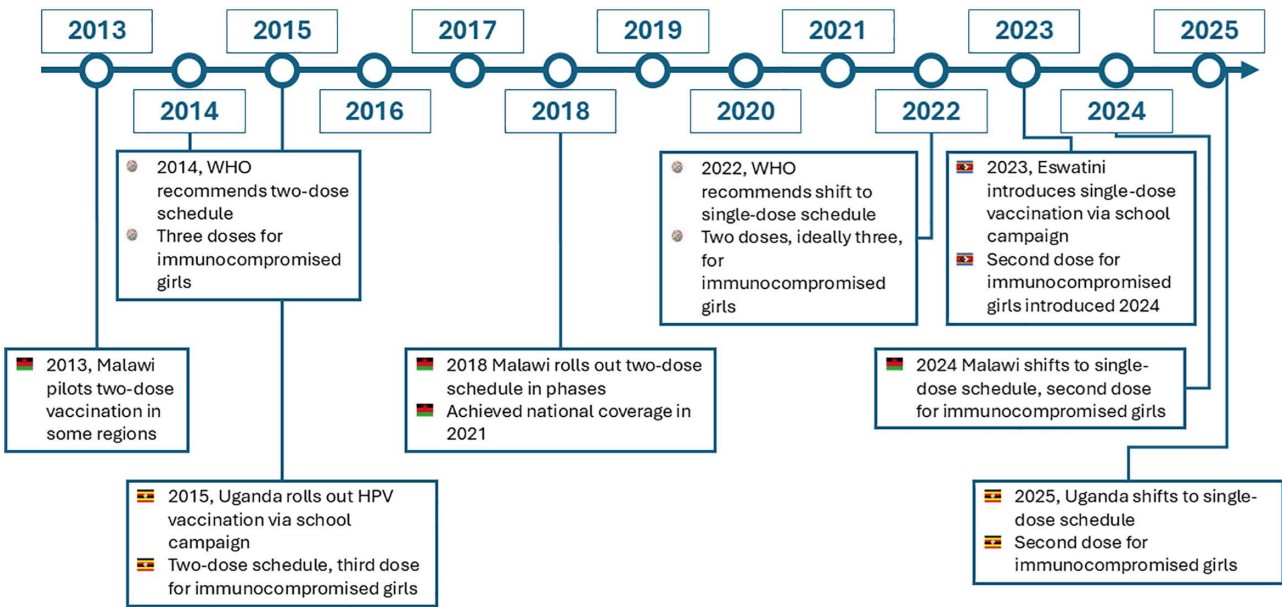

**Fig 1. HPV vaccine introduction timeline in Eswatini, Malawi and Uganda.**

immunocompromised girls. The KII and FGD for Uganda were conducted prior to the announcement of the shift to a single dose.

Participants in the KII and FGD described the strategy for vaccine introduction in each country. Eswatini, Malawi, and Uganda all introduced the vaccine through a campaign-based delivery strategy (Table 4). Following introduction, all three countries have integrated HPV vaccination into routine health services. Malawi continues to deliver the HPV vaccine in schools as a complement to routine vaccination, and Eswatini has plans to conduct a community campaign in 2025. Uganda, in addition to routinely offering the vaccine, also delivers it in communities through Integrated Child Health Days held twice a year.

**Policy and awareness for HPV vaccination for GLHIV and immunocompromised girls**

**There is limited documentation of HPV vaccine policy for GLHIV and immunocompromised girls.** While the dosing schedule for immunocompromised adolescent girls in Eswatini, Malawi, and Uganda is the same in each country—two doses at a six-month interval——the desk review (Table 3) revealed that documentation of this

**Table 4. HPV vaccine introduction and delivery strategies in Eswatini, Malawi, and Uganda.**

|  | Eswatini | Malawi | Uganda |
|---|---|---|---|
| HPV vaccine introduction | 2023 | 2018 | 2015 |
| Single-dose schedule adoption | 2023 | 2024 | 2025 |
| Second dose for immunocompromised girls (year of implementation) | Yes (2024) | Yes (2024) | Yes (2025) |
| Introduction strategy | School campaign | School and community campaign | School campaign |
| Current vaccine strategy | Routine health services | Routine health services, campaigns | Routine health services, outreach |
| Target age (years) | 9-14 | 9 | 10 |
| HPV1 coverage (2023) [41] | 59% | 68% | 99% |

guideline varies by country. The Eswatini Vaccine Integration Framework states that the HPV vaccine should be integrated into health services, including clinics providing specific services for HIV, tuberculosis (TB) and non-communicable diseases (NCDs) [42]; although the guidelines clearly prescribe where the vaccine should be made available, the dose schedule and specific strategies for reaching immunocompromised girls are not documented in any policy or guidelines document. In Uganda, the Director General of Health Services has communicated the dosing schedule for immunocompromised girls via official letter [43], but this has not been formalized as policy. The dose schedule and strategies for reaching immunocompromised girls in Malawi are not documented in any policy or guidelines document.

**Awareness of the dose schedule is inconsistent among stakeholders.** Respondents to the KII and FGD across the study varied in their level of awareness of the dose schedule and strategies to fully vaccinate immunocompromised girls, though there were distinct trends by country.

*Eswatini.* KII respondents in Eswatini reported awareness that girls living with HIV (GLHIV) should receive a second dose of the HPV vaccine, though they lacked clarity on how this is implemented in practice. Some stakeholders recognized that immunocompromised HIV-negative girls also require a second dose, although awareness of this group's needs was lower. Both KII and FGD respondents noted that vaccination strategies for GLHIV have sometimes been intentionally kept private to reduce stigma. However, this confidentiality has contributed to limited awareness among stakeholders, affected girls, and their families regarding the appropriate vaccination process.

*Malawi.* Respondents to both KII and FGDs in Malawi demonstrated low awareness of the recommended dosing schedule for GLHIV, as well as for HIV-negative girls. This did not vary by stakeholder type – awareness was consistently low across government, multilateral, and NGO stakeholders. In FGDs, health workers consistently reported that they administer the same number of doses to all girls regardless of HIV status.

"We are using a strategy that is not differentiating those who are HIV positive or negative" (FGD, Urban Health Workers, Malawi).

Practices varied by facility: one facility reported administering only one dose to all girls, while two other facilities reported providing two doses to all girls.

*Uganda.* Respondents to KII and FGDs in Uganda had low awareness of the dosing schedule for GLHIV. In addition, many expressed opposition to adopting a differentiated schedule, citing concerns that such a policy would stigmatize GLHIV.

"It will increase the stigma and disclosure issues if we focus more on [GLHIV]" (FGD, Urban Health Workers, Uganda).

### Strategies for vaccinating GLHIV

Respondents in KII and FGD reported similar strategies for reaching GLHIV for HPV vaccination across the three countries. In each country, some of the respondents, particularly health workers participating in the FGDs, were able to give examples of ways that GLHIV are identified and given the HPV vaccine. Awareness of strategies was not uniform, however; in each country, some respondents did not know of any strategies to vaccinate GLHIV. The main reported HPV vaccine delivery strategies for GLHIV were identification and delivery through ART clinics and teen clubs.

**ART clinics are a key platform for reaching GLHIV.** Across the three countries, respondents in KIIs and FGDs described ART clinics as an important entry point for vaccinating GLHIV, though practices and awareness varied. In Malawi, FGD participants reported offering HPV vaccination to girls attending ART clinics, although vaccination records do not capture HIV status. In Uganda, KII respondents cited ART clinics that offer youth-friendly services and linkage facilitators to connect GLHIV to health care. In an FGD, a health worker described scheduling vaccine appointments

during ART refill visits - though overall it was not clear whether HPV vaccination is routinely offered in these clinics. In Eswatini, Ministry of Health (MOH) personnel and health care providers interviewed in KIIs and FGDs described more established practices, including training and supportive supervision for health workers in ART clinics to administer the HPV vaccine. They also reported multiple approaches for identifying eligible girls: some are offered the vaccine during their regular ART visits, others through coordination with Teen Clubs, and some through the Client Management Information System (CMIS).

Respondents reported advantages and disadvantages to ART clinic-based vaccination. Respondents in Eswatini said a girl's positive relationship with her healthcare provider at the ART clinic helps with trust and uptake.

"[ART nurses] already have a strong relationship with the client, which encourages them to vaccinate as opposed to them being told to vaccinate by outsiders" (KII, NGO staff, Eswatini).

However, respondents across countries stated that clinics lack appropriate cold storage for vaccines. In Eswatini, KII respondents who worked across clinics described decentralized solutions specific to clinics, including storing vaccines in the EPI department, temporarily storing vaccines in the maternity unit on ART clinic days, and storing vaccines in cooler boxes during ART clinic hours.

**Teen Clubs are a platform for identifying and vaccinating GLHIV.** Respondents across Eswatini, Malawi, and Uganda described HPV vaccination in Teen Clubs, which are support groups for adolescents living with HIV (ALHIV) and the place where many adolescents receive their ART. Health care providers in an FGD in Malawi described this process in detail. These providers delivered messaging and vaccines at Teen Club meetings with three key steps: conducting health talks about the HPV vaccine with information targeted specifically to GLHIV, framing the vaccine as a routine preventive measure, and administering the vaccine during the same session. The providers noted that Teen Clubs offer other adolescent-friendly health services such as family planning services, and that meetings are an opportunity to conduct concurrent health discussions with parents to promote vaccine uptake. This was echoed in Eswatini, where a CHW said that:

"[GLHIV's] parents were willing to listen to us because of the relationship we have in correspondence with their children, the trust they have in us" (FGD, Eswatini).

Respondents noted key advantages to vaccine delivery in Teen Clubs, noting that this is an entry point where GLHIV have strong peer support and face little risk of stigma. Challenges to delivery through Teen Clubs were also noted, namely a lack of vaccine cold storage in meeting venues, irregular Teen Club meeting dates, and a lack of reliable transportation for some GLHIV to attend meetings.

**Community outreach and other approaches support identification of GLHIV.** Key informants in Malawi and Eswatini noted limited use of community outreach to identify and vaccinate GLHIV. KII respondents in Malawi cited examples of CHWs conducting door-to-door outreach to identify eligible girls for vaccination. In Eswatini, respondents noted that CHWs speak to GLHIV in their homes to educate them about the need for a second dose.

**Record-keeping is carefully managed to avoid stigma.** In Eswatini, respondents to KII and FGD noted that the second dose of HPV vaccine is recorded in their HIV patient card rather than their HPV vaccination booklet to avoid stigma.

**Fewer strategies exist for immunocompromised HIV-negative girls.** Participants in KII and FGD across the study countries focused on strategies for vaccinating GLHIV; in Malawi and Uganda, no participants mentioned vaccinating HIV-negative immunocompromised girls. Some KII respondents in Eswatini mentioned integration of HPV vaccination in other areas of the hospital for girls who have other immunocompromising health conditions, such as cancer or diabetes.

**Barriers for HPV vaccination among GLHIV.** HIV-related stigma was reported as an important barrier to HPV vaccination in FGD and KII across all three countries. Difficulty speaking about HIV status leads to lower health worker knowledge about differentiated dosing schedules and difficulty identifying GLHIV. It also leads to lower levels of awareness amongst parents of GLHIV that their daughters require a second dose of the vaccine.

**Stigma leads to challenges in disclosing HIV status, complicating access to differentiated care.** Key informants cited examples of stigma-driven challenges related to HIV disclosure affecting vaccination. Key informants across all three countries shared the need for specific strategies to sensitize and vaccinate GLHIV to avoid stigma from others and accidental disclosure of HIV status to GLHIV and their peers. Several stakeholders commented:

"We want to avoid announcing through the radio that 'all those living with HIV or immunocompromised must come for vaccinations for second dose' to avoid shaming and stigmatization" (KII, Government, Eswatini).

"[Adolescents] ask each other 'who will get the second dose' as they know what kind of adolescents get a second dose, and that is stigmatization, and it starts there" (FGD, Nurses, Eswatini).

KIIs in Eswatini revealed that not all GLHIV are told their HIV status by 9 years, the age of vaccine initiation. Administering two doses of the vaccine to GLHIV requires careful explanation to avoid inadvertent disclosure. This barrier is even more defined in Uganda, where key informants reported that the age of HIV disclosure is 13 years, whereas the age of HPV vaccination is 10 years.

"We need to be careful about disclosure, because that is a very sensitive thing and most children that age do not know their status" (KII, Government, Uganda).

**GLHIV require tailored health communications around HPV vaccination.** A lack of tailored messaging around HPV vaccination for GLHIV was reported as another important barrier. Key informants in Eswatini reported that GLHIV and their parents often do not trust health messaging unless it comes from their child's nurse at the ART clinic. They also said that GLHIV struggle with treatment fatigue and may resist vaccination due to a feeling that they are constantly subject to new treatments and initiatives. This lack of specific health messaging for GLHIV also leads to low awareness for GLHIV and their parents of their need for a second dose of the vaccine, limiting demand.

**Weak HIV-specific data makes tracking HPV vaccination for GLHIV difficult.** Health system weaknesses were another commonly reported barrier. In Malawi and Uganda, KIIs reported that HMIS and paper tools for HPV vaccination do not indicate HIV status, leading to an inability to monitor vaccination for GLHIV. Health workers in FGDs in Malawi and Uganda also reported stockouts for paper tracking tools for HPV vaccination in health facilities and for CHWs. These respondents highlighted gaps that prevent identification of GLHIV who require a second vaccine dose, stop GLHIV and their parents from receiving relevant information about their health needs, and lead to poor tracking of vaccine coverage amongst GLHIV.

**GLHIV also experience vaccine barriers not directly related to HIV.** Across KIIs and FGDs in all three countries, the most commonly reported barrier to vaccination was stockouts due to inconsistent supply chain, a challenge that affects GLHIV and HIV-negative girls alike. Key informants reported that lack of clear policy for HPV vaccination, lack of coordination across health system departments, and weak financing were also key barriers affecting GLHIV in Eswatini, Malawi, and Uganda. In Malawi, FGDs demonstrated low provider knowledge about the appropriate dosing schedule as another important barrier.

Community misconceptions and misinformation as a barrier to HPV vaccination was also commonly reported in KII and FGD across the three countries. Respondents say that many people believe the vaccine is a method of contraception, a cause of infertility, or even a way to kill adolescent girls. The belief that the vaccine is satanic or a form of devil worship

was also reported in all three countries. In Malawi and Uganda, FGD respondents also reported that parents of GLHIV fear that the vaccine could harm their child or reduce the effectiveness of their ART. Health workers in Uganda said:

> "[Some people] think that HPV vaccine can lead to cancer for the HIV positives due to their reduced immunity" (FGD, Health workers, Uganda).

In Uganda, respondents in both KII and FGD, including health workers, confuse HIV and HPV, leading to confusion around what the HPV vaccine is for.

**Enablers for HPV vaccination among GLHIV.** HIV-specific services for adolescent girls, including established ART clinics and teen clubs, were cited by key informants and FGD participants as a key enabling platform in Eswatini, Malawi, and Uganda. Respondents indicated that differentiated care for GLHIV is already common in Eswatini and has potential for scale-up in Malawi and Uganda. The desk review gave evidence that health systems in Eswatini and Uganda have guidelines to integrate HPV vaccination, and in Eswatini this guidance specifically speaks to HIV clinics. Respondents working in the Eswatini health system reported additional support for health workers in the form of training on HPV vaccine delivery to GLHIV and data systems that disaggregate HPV vaccination by HIV status.

Respondents in Eswatini and Uganda cited positive relationships between GLHIV and their health care providers as a promoter of vaccine uptake.

> "When [GLHIV] trust a nurse, that means they know whatever he/she says about their health is truthful and they are easy to open up about their chronic health and also receive the vaccine" (KII, Government, Eswatini).

In Malawi, community outreach and peer support were also named as enablers. In Uganda, family support groups, peer support strategies, and adolescent clinics were also named as existing structures that could support HPV vaccination for GLHIV.

A commonly named enabler for HPV vaccination in all three countries was community awareness of cervical cancer. In Eswatini and Malawi, stakeholders credited effective communications campaigns with creating demand for cervical cancer prevention through HPV vaccination. In all three countries, respondents said that a fear of cervical cancer and a desire to prevent it acts as a key driver in vaccine uptake. Furthermore, health system stakeholders that participated in the KII indicated strong knowledge about and commitment to HPV vaccination.

## Discussion

This study describes the policies, strategies, barriers, and enablers for vaccinating immunocompromised girls, with a focus on GLHIV, in Eswatini, Malawi, and Uganda. We used KIIs, FGDs, and a desk review to improve understanding of the policy landscape, the extent to which each country implements strategies to deliver the vaccine to these girls, and the associated barriers and enablers [21,24,25]. We found that all three countries are aligned with WHO recommendations to offer a single dose HPV vaccine schedule and a second dose for immunocompromised girls and have existing strategies for vaccinating GLHIV. These promising strategies mainly rely on identification of eligible GLHIV and delivery of HPV vaccines through two key platforms that serve GLHIV—ART clinics and teen clubs. Nevertheless, none of the countries have formally scaled strategies to vaccinate GLHIV, and stakeholders consistently have lower knowledge about vaccinating immunocompromised girls compared to immunocompetent girls. Stigma, weak HIV-focused vaccine delivery, and inadequate data systems act as barriers for fully vaccinating GLHIV. Enablers include ongoing HIV-specific care, positive patient-caregiver relationships among GLHIV, and community awareness of cervical cancer. The findings can be interpreted through the PRISM framework to further understand how intervention characteristics, external context, implementation infrastructure, and stakeholder perspectives influence implementation outcomes [39,40].

HPV vaccination policy influenced implementation across Eswatini, Malawi, and Uganda. While all three countries align with the WHO recommendation for a two-dose HPV schedule for immunocompromised girls, the clarity and detail of guidance of the policy varies substantially. Eswatini's Vaccine Integration Framework references incorporating HPV vaccination into services for HIV, TB, and non-communicable diseases, creating an enabling entry point, but lacks specific guidance on dosing schedules and implementation strategies [42]. Uganda has issued an official letter communicating the dosing schedule for GLHIV, but this lacks the authority of a national policy [43]. In contrast, Malawi has no written policy documents. These gaps have led to uncertainty and inconsistent delivery among frontline health workers [23,44] and limited resource allocation for the infrastructure, training, and supervision needed for effective implementation. In systems where HPV vaccination is delivered through multiple platforms—schools, health clinics, and outreach programs—lack of a clear, unified policy exacerbates confusion and opens the door to inequities in access. The challenge is further compounded for GLHIV, who may experience stigma or late HIV disclosure that interferes with their ability to access a tailored vaccination schedule. Prior research echoes these findings, showing that WHO guidance, when adopted, must be translated into clear, operational policy and communicated throughout all levels of the health system [9,21,26].

This study revealed how both barriers and enablers relate to the implementation and sustainability infrastructure pillar of the PRISM framework. Findings highlighting infrastructure challenges, namely those related to frequent vaccine and tool stockouts, echo existing literature that demonstrates weak infrastructure and health system constraints as barriers regardless of HIV or immunocompromised status [44,45]. However, the lack of infrastructure tailored to GLHIV, including a lack of HIV-disaggregated data in Malawi and Uganda and limited cold storage at ART clinics across all study countries demonstrates how infrastructure magnifies inequality. These challenges constrain identification and follow up of GLHIV. At the same time, enabling infrastructures are visible: Eswatini's Client Management Information System (CMIS) provides disaggregated data to track vaccination among GLHIV, while training and supportive supervision build health worker capacity. ART clinics and Teen Clubs serve as existing platforms that could integrate HPV vaccination into ongoing HIV care.

The knowledge, attitudes, and behaviors of health workers, community members, and GLHIV are clearly linked to HPV vaccine implementation. Providers in Malawi and Uganda lacked awareness of differentiated dosing schedules, and some in Uganda opposed them out of concern for stigma. By contrast, Eswatini's providers, supported by training and supervision, described confidence in delivering the HPV vaccine and even developed creative solutions to overcome challenges. For GLHIV, respondents echoed the findings of Huff et al. [46], who found that individuals with HIV are often unaware that HPV vaccination is recommended or beneficial for them. Stakeholders also noted treatment fatigue as a barrier for GLHIV. However, this study showed that the long-term relationship between GLHIV and their ART provider is an enabler for vaccine uptake. Peer support was also seen here and elsewhere as an enabler of HPV vaccination among GLHIV [27]. Community awareness of the risks of cervical cancer, previously documented in similar studies, is another critical enabler related to the characteristics of implementers and recipients [23,47].

Despite policy and operational gaps, this study identified promising strategies to deliver the HPV vaccine to GLHIV in all three countries. The core strategies are Teen Clubs and adolescent-friendly ART clinics, which serve as key touchpoints for ongoing HIV care [20,27]. These settings offer trusted relationships between health workers and adolescent girls and have been successfully leveraged to deliver HPV vaccination in some contexts. In Malawi, for example, providers introduce the vaccine through tailored health talks at Teen Clubs, administer it during ART refill visits, and engage peer mentors to encourage uptake. Integration of other adolescent health services, such as family planning, further normalizes vaccination and increases service acceptability. These findings align with other literature showing the effectiveness of peer-led, adolescent-centered HIV care in promoting vaccine uptake [27]. In Eswatini, HPV vaccination is delivered through HIV clinics within a package of structured training for ART clinic staff, supportive supervision, and data tools like the CMIS. Health facilities have adapted to logistical issues by using cooler boxes or coordinating with maternity wards or EPI units to store vaccines on ART clinic days. These integrated delivery models demonstrate the feasibility of combining

services and may offer replicable models for other countries with high burdens of HIV and cervical cancer. This type of service integration is likely to be feasible across other high-burden HIV and cervical cancer countries and is currently being tested in Zambia, but may require tailoring to country contexts and sufficient resource allocation for scale-up and sustainability [48].

### Study limitations and strengths

This study offers several key strengths. First, it provides, to our knowledge, the first multi-country evidence addressing HPV vaccination in GLHIV, addressing a gap in the published literature and supplying decision-makers with urgently needed information. Second, the study draws on a rich variety of perspectives across national and subnational levels, including policymakers, implementing partners, and frontline providers, supplemented by a review of key policy documents. In-country research team members with direct roles in HPV and HIV programs enhanced stakeholder identification, contextual insight, and the validity of interpretations. Finally, the use of analysis approaches such as summary notes and AI-assisted analysis allowed a small team of programmatic staff to work efficiently with limited resources and still produce timely, actionable findings to inform policy decisions and produce evidence for publication.

Notwithstanding its strengths, the study is constrained by several methodological and contextual limitations. The sampling approach was guided by convenience, particularly in subnational geographies, which may introduce selection bias and limit the representativeness of findings within each country. The data collection and analysis did not include program budgets or financial records, limiting our understanding of the role of resource constraints in vaccinating GLHIV. Additionally, our analysis of synthesized interview notes rather than full transcripts, together with the use of ChatGPT for initial thematic exploration, may have constrained the depth and subtlety of the analysis, though these limitations were mitigated by restricting AI input to exploratory guidance.

**Implications for policy and practice.** The findings of this study point to several critical actions needed to strengthen HPV vaccine delivery for immunocompromised girls, particularly GLHIV, in Eswatini, Malawi, and Uganda. These findings are likely applicable to other health systems, particularly those in LMICs with a high burden of HIV among adolescent girls. While national adoption of WHO's recommended two-dose schedule for immunocompromised girls represents important progress, effective implementation will require intentional, equity-focused design and operational planning. This study also shares insights into barriers and enablers for life-course vaccination and differential service delivery.

### Develop and disseminate HPV vaccination policies for immunocompromised girls

Countries would benefit from the development and dissemination of clear, national policy documents that explicitly define the HPV dosing schedule and delivery strategies for immunocompromised girls, including GLHIV. Countries may consider developing guidelines collaboratively with personnel from departments that focus on immunization, HIV, adolescents, and girls. Guidelines that describe delivery strategies across service points—particularly HIV clinics, Teen Clubs, and adolescent-friendly services—and that are supported by accompanying training, supervision tools, and job aids for frontline providers, would be most effective. These actions could translate to improved service consistency and health worker confidence [9,23,26,49].

### Strengthen and scale up existing HIV service delivery mechanisms

This study identified successful existing practices that could be scaled. HIV care infrastructure, especially Teen Clubs and ART clinics, provide a high-potential platform for integrated HPV vaccine delivery. Leveraging peer mentorship, synchronized vaccine and ART delivery, and use of adolescent-friendly messaging may also be beneficial, though must be adapted to national and subnational contexts. Infrastructure would benefit through implementation of innovative strategies to address cold chain limitations in HIV clinics. Evidence indicates that scheduled vaccine delivery days combined

with cooler boxes or interdepartmental coordination can improve vaccine storage and availability. Health worker training on delivering HPV vaccination alongside routine HIV services could reduce missed opportunities to fully vaccinate this high-risk group. National Health Systems may benefit from structured learning exchanges to share best practices, identify gaps, and develop contextually appropriate adaptations [27,42]

### Strengthen HPV data and monitoring systems for GLHIV

Evidence from Malawi and Uganda highlights the opportunity to adapt data systems to capture, disaggregate, and report on vaccine coverage for immunocompromised groups. Improved data, achieved through integration of HPV vaccination into ART registers, patient-held cards, and digital health tools, will identify gaps and hold health systems accountable [3,17,44]. Stakeholders in Eswatini discussed the advantages of the CMIS, which incorporates SMS reminder systems. These reminders, also demonstrated in Malawi, could improve second-dose completion if funded effectively [50,51].

### Reach HIV-negative immunocompromised and other marginalized girls

While GLHIV are a well-defined target population within the health system, HIV-negative immunocompromised girls—such as those with cancer, diabetes, or autoimmune conditions—may be harder to identify and vaccinate. In alignment with the principles of universal health care, countries could develop strategies to ensure equitable access to care for these girls [52]. Approaches may identify these girls through specialized care units or through linkage with NCD programs as advised in Eswatini's Vaccine Integration Framework [42]. Additional outreach strategies would benefit other marginalized populations, including girls who live far from clinics, are newly diagnosed with HIV, migrate for work, or are not currently in HIV care. These strategies are most effective when developed in collaboration with organizations already working with these groups and based on principles of equity and community engagement [21].

### Promote stigma-sensitive, gender-aware community engagement

Stigma around HIV and gender norms were identified as central barriers to HPV vaccination for GLHIV. National programs could address this by co-designing communication strategies with adolescents, caregivers, peer mentors, and healthcare providers to ensure messaging is accurate, inclusive, and sensitive to privacy concerns. This includes carefully framing HPV vaccination for GLHIV as part of routine preventive care rather than a separate, stigmatized intervention. Building on strong community awareness of cervical cancer prevention—an enabler identified across countries—can help shift the narrative toward protection and empowerment [46,47,53].

### Conclusion

This study reveals that while HPV vaccine strategies for immunocompromised girls exist in principle, intentional design and implementation are largely lacking. Countries can leverage existing HIV service platforms, adolescent-friendly infrastructure, and cervical cancer awareness to close this equity gap. Expanded Program on Immunization (EPI) and HIV programs should collectively work together to design targeted vaccination strategies for GLHIV. Formalizing, resourcing, and monitoring strategies for vaccinating immunocompromised girls is essential for cervical cancer prevention in high HIV-burden settings.

### Supporting information

**S1 Text. Interview Guides.**
(DOCX)

**S2 Text. Inclusivity in global research Questionnaire.**
(DOCX)

## Acknowledgments

We thank the Ministries of Health in Eswatini, Malawi, and Uganda for their support and collaboration. We also acknowledge the contributions of field researchers and data collection teams in all three countries, as well as the study participants—including health workers, policy-makers, and community leaders—whose insights informed this work. In Malawi, we thank Tuweni Chumachapera and Dr. Henry Phiri from the Ministry of Health, and Andrews Gunda from the Clinton Health Access Initiative (CHAI). In Eswatini, we thank Nyasatu Ntshalintshali and Nomfundo Mncina from CHAI. We also acknowledge Jessica Gu, Shadrack Mngemane, and Laure Anais Zultak from CHAI's Global Vaccines Delivery team. ChatGPT (OpenAI, April 2024 version) was used to assist with qualitative data analysis and editorial support for the first draft of the manuscript. All AI-assisted content was reviewed and refined by the study team.

## Author contributions

**Conceptualization:** Tosin F. Ajayi, Immaculate Ampeire, Michael Baganizi, Mike N. Chisema, Bridget C. Griffith, Lorraine Kabunga, Thuli Magagula, Lisa-Rufaro Marowa, Akachi E. Mbogu, Nobuhle Mthethwa, Stella Namutebi, Timothy Tchereni, Frehiwot Birhanu, Xolisiwe Dlamini, Fredrick Luwaga, Sonali Patel.

**Data curation:** Emily E. Crawford.

**Formal analysis:** Emily E. Crawford.

**Funding acquisition:** Tosin F. Ajayi.

**Investigation:** Mike N. Chisema, Stella Namutebi, Bhekiwe Shongwe, Timothy Tchereni, Frehiwot Birhanu, Fredrick Luwaga.

**Methodology:** Akachi E. Mbogu.

**Project administration:** Bhekiwe Shongwe, Sonali Patel.

**Supervision:** Timothy Tchereni, Sonali Patel.

**Validation:** Lisa-Rufaro Marowa, Stella Namutebi, Bhekiwe Shongwe, Timothy Tchereni.

**Visualization:** Emily E. Crawford.

**Writing – original draft:** Emily E. Crawford.

**Writing – review & editing:** Emily E. Crawford, Tosin F. Ajayi, Immaculate Ampeire, Michael Baganizi, Mike N. Chisema, Bridget C. Griffith, Lorraine Kabunga, Thuli Magagula, Lisa-Rufaro Marowa, Akachi E. Mbogu, Nobuhle Mthethwa, Stella Namutebi, Bhekiwe Shongwe, Timothy Tchereni, Frehiwot Birhanu, Xolisiwe Dlamini, Fredrick Luwaga, Sonali Patel.

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
