## [Decision Letter · Decision Letter 0]

25 Jul 2025

PGPH-D-25-01096

Policies and strategies for HPV vaccination schedule completion in immunocompromised girls, including girls living with HIV: Qualitative insights from Eswatini, Malawi, and Uganda

Dear Dr. Patel,

Thank you for submitting your manuscript to PLOS Global Public Health. After careful consideration, we feel that it has merit but does not fully meet PLOS Global Public Health’s publication criteria as it currently stands. Therefore, we invite you to submit a revised version of the manuscript that addresses the points raised during the review process.

Please do take note of the Editor's and Reviewers' comments listed within this email and in attachments.

A rebuttal letter that responds to each point raised by the editor and reviewers. You should upload this letter as a separate file labeled 'Response to Reviewers'.A marked-up copy of your manuscript that highlights changes made to the original version. You should upload this as a separate file labeled 'Revised Manuscript with Track Changes'.An unmarked version of your revised paper without tracked changes. You should upload this as a separate file labeled 'Manuscript'.

We look forward to receiving your revised manuscript.

Kind regards,

Edina Amponsah-Dacosta, Ph.D., MPH

Academic Editor

Journal Requirements:

2. In the online submission form, you indicated that The qualitative data for this study were collected using structured note-taking matrices rather than full transcripts. These de-identified matrices contain potentially sensitive information and are not publicly available. However, de-identified data for each country included in the study may be made available upon reasonable request to the corresponding author and subject to approval from the relevant national ethics committee in each country.

3. Uploaded as supplementary information.

Reviewers' comments:

Reviewer's Responses to Questions

**Comments to the Author**

1. Does this manuscript meet PLOS Global Public Health’s publication criteria ? Is the manuscript technically sound, and do the data support the conclusions? The manuscript must describe methodologically and ethically rigorous research with conclusions that are appropriately drawn based on the data presented.

Reviewer #1: Yes

Reviewer #2: Yes

Reviewer #3: Partly

Reviewer #4: Partly

2. Has the statistical analysis been performed appropriately and rigorously?

Reviewer #1: No

Reviewer #2: N/A

Reviewer #3: No

Reviewer #4: No

3. Have the authors made all data underlying the findings in their manuscript fully available (please refer to the Data Availability Statement at the start of the manuscript PDF file)?

Reviewer #1: Yes

Reviewer #2: Yes

Reviewer #3: No

Reviewer #4: Yes

4. Is the manuscript presented in an intelligible fashion and written in standard English?

Reviewer #1: Yes

Reviewer #2: Yes

Reviewer #3: Yes

Reviewer #4: Yes

5. Review Comments to the Author

Reviewer #1: Thank you for this manuscript, it is quite insightful into the challenges with HPV vaccine uptake in young women living with HIV.

Please see a few comments/clarifications requested below:

1. The acronym LMIC is first used in the background/introduction without spelling out in full (line 67)

2. The paper refers to immunocompromised girls, including those living with HIV, but focuses exclusively on data related to girls living with HIV. If the data collected were only from this group, the authors could consider removing references to the other immunocompromised girls and including them in the Discussion section to highlight how the findings could also inform outreach to these populations.

3. The authors have communicated well that the qualitative data for this study were collected using structured note-taking matrices rather than full transcripts.

• Can we access/see the interview guides? Please confirm if they were submitted.

4. The section on HPV vaccine introduction seems misplaced under the RESULTS section. It should be part of the background (line 186)

5. Line 226, please differentiate in your results responses from stakeholders and health care workers.

6. KII were conducted with stakeholders from the Ministry of Health, other government entities, and implementing partners; FGDs were conducted with health care workers and Community health workers

• Is there a way it can come out clearly from the results section who the respondents were based on their roles in vaccine delivery? It is not immediately clear in some sections who the respondents were

7. If the strategies for vaccination are somewhat similar across the 3 countries, can these results be given under one subheading?

8. All data was analyzed by thematic analysis using the note-taking matrices. Qualitative data analysis was assisted by ChatGPT (OpenAI, April 2024 version), a large language model, which supported the thematic analysis and synthesis of the data. Raw data was shared with ChatGPT in thematic sections, and the model was prompted to carry out thematic analysis

a. Thank you for disclosing this. I am not cognisant on use of ChatGPT as the sole coder or analyst of raw qualitative data Please attach any IRB or regulatory document from your institution regarding use of AI tools for analysing raw data.

b. Were there any other steps in the statistical analysis? Was there a codebook? It would help to explain the researchers' oversight in refining the themes.

9. If possible can you add a figure of the PRISM framework used and show in an illustration the contextual factors used to address the inequity in access to HPV vaccine highlighted in the paper?

Reviewer #2: The article by Crawford et al. presents a study of HPV vaccination in immunocompromised girls in Eswatini, Uganda, and Malawi, which is a timely and valuable contribution, particularly in the context of immunocompromised populations in sub-Saharan Africa—a population often underrepresented in global vaccine delivery studies. The manuscript is well-structured and clearly written, and the study methodology is consistent. However, the manuscript is currently too lengthy in parts, and the key messages risk being diluted. The authors might consider summarizing repetitive points.

Reviewer #3: 1. Title: The title could be improved by removing redundancies, specifically the reference to both "immunocompromised girls" and "girls living with HIV," given that HIV is the primary driver of immunosuppression in this study population. Additionally, to better reflect the focus and methodology, consider rephrasing to specify that these are stakeholder perspectives on policies and strategies. For example: “Stakeholder perspectives on policies and strategies for HPV vaccination schedule completion among girls living with HIV: Qualitative insights from Eswatini, Malawi, and Uganda.”

2. Ethical approval: The manuscript states that the study received ethical approval from the Clinton Health Access Initiative’s (CHAI) internal Scientific and Ethical Review Committee. Kindly provide the reference number for this approval. Furthermore, please clarify whether this committee is accredited by the Uganda National Council for Science and Technology (UNCST). Based on the publicly available list of accredited committees as of February 2024 https://www.uncst.go.ug/files/downloads/ACCREDITED%20RESEARCH%20ETHICS%20COMMITTEE%20IN%20UGANDA%20AS%20AT%20FEBRUARY%202024.pdf, CHAI's committee does not appear to be listed.

3. Use of ChatGPT in qualitative analysis: It is notable that ChatGPT (OpenAI, April 2024 version) was used to assist in qualitative data analysis, which is not a conventional tool for such work. Most qualitative research typically employs software such as NVivo or Atlas.ti. Please clarify:

• What was the rationale for using ChatGPT instead?

• How were prompts or commands phrased to maintain analytical rigor and consistency?

• How were issues of data privacy, participant consent, and transparency addressed in the context of AI-assisted analysis?

This information is crucial given the novelty of AI use in this context and the ethical considerations it raises.

4. Presentation of participant voices: The manuscript would benefit from the inclusion of illustrative participant quotes under each identified theme. This is standard in qualitative research to provide context and depth. For example, under the theme “Stigma leads to challenges in disclosing HIV status, complicating access to differentiated care,” the current narrative does not make clear who expressed these views. Were the participants themselves WLHIV/GLHIV, or speaking about them? Including at least 3–4 relevant quotes would enhance the trustworthiness and authenticity of the findings.

5. Discussion section clarity: The discussion section would benefit from improved structure. For example, lines 518 to 575 appear to describe key implications for policy and practice. If so, this should be explicitly stated by introducing the subsection (e.g., “This study provides the following policy and practice implications…”). This will guide the reader and improve overall clarity.

6. Results and discussion subheadings: Both the results and discussion sections are currently divided into subsections with subheadings. However, these subheadings are many and some how similar. Making them less and more concise would significantly improve readability and comprehension.

7. Figure 1: Please reference Figure 1 in the main text, as it currently appears unmentioned. It provides useful information and should be integrated more clearly into the narrative.

8. References

Several references require correction:

• Reference 20: The provided link does not lead to the WHO Weekly Epidemiological Record as indicated.

• Reference 21: The citation for PLOS Global Public Health is inconsistently formatted:

“PLOS 674 Global Public Health. 2024;4: e0003931. doi:10.1371/JOURNAL.PGPH.0003931”

• Reference 24 is incomplete.

Please revise these and ensure all references conform to the Vancouver reference style, as recommended by the journal.

9. Acknowledgements section

The purpose of the acknowledgements section is to recognize individuals who contributed to the work but do not meet authorship criteria (e.g., field researchers, data collectors, analysts). It is not standard practice to include study participants in this section. Kindly revise accordingly by removing the mention of participants and instead acknowledging the specific contributions of the field or data collection teams

Reviewer #4: This study is highly policy relevant. Please consider doing justice to such a study by improving sentence clarity, tightening your arguments, reporting/synthesizing the results better and making the discussion more analytical rather than descriptive.

6. PLOS authors have the option to publish the peer review history of their article (what does this mean? ). If published, this will include your full peer review and any attached files.

**Do you want your identity to be public for this peer review?** For information about this choice, including consent withdrawal, please see our Privacy Policy .

Reviewer #1: No

Reviewer #2: No

Reviewer #3: **Yes:** Ludoviko Zirimenya

Reviewer #4: No

---

## [Decision Letter · Decision Letter 1]

2 Nov 2025

PGPH-D-25-01096R1

Policies and strategies for HPV vaccination schedule completion in immunocompromised girls, including girls living with HIV: Qualitative insights from Eswatini, Malawi, and Uganda

Dear Dr. Patel,

Thank you for submitting your revised manuscript to PLOS Global Public Health. After careful consideration, we feel that it much improved but does not fully meet PLOS Global Public Health’s publication criteria as yet. Therefore, we invite you to submit a revised version of the manuscript that addresses the points raised during the review process.

We look forward to receiving your revised manuscript.

Kind regards,

Edina Amponsah-Dacosta, Ph.D., MPH

Academic Editor

Journal Requirements:

Reviewers' comments:

Reviewer's Responses to Questions

**Comments to the Author**

1. If the authors have adequately addressed your comments raised in a previous round of review and you feel that this manuscript is now acceptable for publication, you may indicate that here to bypass the “Comments to the Author” section, enter your conflict of interest statement in the “Confidential to Editor” section, and submit your "Accept" recommendation.

Reviewer #1: All comments have been addressed

Reviewer #2: All comments have been addressed

Reviewer #3: (No Response)

Reviewer #4: (No Response)

2. Does this manuscript meet PLOS Global Public Health’s publication criteria ? Is the manuscript technically sound, and do the data support the conclusions? The manuscript must describe methodologically and ethically rigorous research with conclusions that are appropriately drawn based on the data presented.

Reviewer #1: Yes

Reviewer #2: Yes

Reviewer #3: Yes

Reviewer #4: Partly

3. Has the statistical analysis been performed appropriately and rigorously?

Reviewer #1: Yes

Reviewer #2: Yes

Reviewer #3: Yes

Reviewer #4: Yes

4. Have the authors made all data underlying the findings in their manuscript fully available (please refer to the Data Availability Statement at the start of the manuscript PDF file)?

Reviewer #1: Yes

Reviewer #2: Yes

Reviewer #3: (No Response)

Reviewer #4: Yes

5. Is the manuscript presented in an intelligible fashion and written in standard English?

Reviewer #1: Yes

Reviewer #2: Yes

Reviewer #3: Yes

Reviewer #4: Yes

6. Review Comments to the Author

Reviewer #1: The topic addressed is important as most countries are switching to single-dose for young women. Thank you for focusing on LMIC delivery challenges. I have a minor comment:

• Use of ChatGPT for Qualitative Data to help with thematic analysis. This transparency is appreciated 

Reviewer #2: The manuscript PGPH-D-25-01096R1 is suitable for publication in PLOS Global Public Health. It presents a well-structured and policy-relevant qualitative study on HPV vaccination strategies for immunocompromised girls, including those living with HIV, in Eswatini, Malawi, and Uganda. The authors have addressed previous reviewer and editorial comments comprehensively, improving the clarity, thematic synthesis, and analytical depth of the manuscript. The use of the PRISM framework, integration of stakeholder perspectives, and actionable policy recommendations enhance the manuscript's contribution to global health literature. I recommend acceptance.

Reviewer #3: One pending issue that needs further clarification, is what ethics committee reviewed and approved this work in Uganda, it is mentioned that it was internally reviewed by the Clinton Health Access Initiative’s internal Scientific and Ethical Review Committee, is this the one that approved the study?

Reviewer #4: The manuscript has significantly improved. Thank you for making the requisite revisions. However, several raised earlier remain unaddressed:

**INTRODUCTION**

The problem statement needs strengthening. What prior research exists, what didn’t they address that your study aims to address?

**METHODS**

Thematic analysis approach is underspecified; No mention of whether **inductive or deductive coding** was used. No mention of **steps followed** (e.g., familiarization, coding, theme generation).

In the current **methods section****, there is no mention of trustworthiness** , which is a crucial component in qualitative research to demonstrate **credibility, dependability, confirmability, and transferability** of findings.

**RESULTS**

You mention “other government entities” as one of the stakeholders. Who are these?

Is there a reason why supporting quotes from the FDGs do not feature in your results section?

The results section can be synthesized better.

Instead of describing each country’s results separately, consider synthesizing across contextsRestructure around clear sub-themes that point directly to your core objectivesCondense barriers/enablers sectionHave a comparative summary section at the end

**DISCUSSION**

Please use the first paragraph of the discussion to report 3-4 key findings that will be discussed subsequently.

I see in the paragraph of this section you go into a gap analysis. Not the place. This ought to have come out explicitly in the last half of the introduction.

The discussion needs a major revamp. This section is still re-describing findings (“respondents said…”, “this study identified…”) instead of interpreting and analyzing them. The reader is left asking “*so what?”*   i.e., what do these findings mean for policy, implementation, or theory?

The structure (“Cross-country gaps”, “Barriers”, “Enablers”, “Promising strategies”) is fine, but transitions between them are abrupt and there’s no synthesis paragraph drawing them together. Please consider adding linking statements and a short integrative paragraph at the end of each subsection to show progression.

You cite well, but many citations are dropped in without explaining how they extend, support, or contrast with your findings.

I only see a limitations section. Provide a summary of the strengths of this study.

The policy implication section can be shortened for clarity and written in prose. No need for the subtitles. Also be intentional with linking specific policy recommendations with specific country findings otherwise they appear generic.

7. PLOS authors have the option to publish the peer review history of their article (what does this mean? ). If published, this will include your full peer review and any attached files.

**Do you want your identity to be public for this peer review?** For information about this choice, including consent withdrawal, please see our Privacy Policy .

Reviewer #1: No

Reviewer #2: **Yes:** Phelele Bhengu

Reviewer #3: **Yes:** Ludoviko Zirimenya

Reviewer #4: No

 Figure Resubmissions:

---

## [Editor Report · Decision Letter 2]

5 Jan 2026

Policies and strategies for HPV vaccination schedule completion in immunocompromised girls, including girls living with HIV: Qualitative insights from Eswatini, Malawi, and Uganda

PGPH-D-25-01096R2

Dear Ms Patel,

We are pleased to inform you that your manuscript 'Policies and strategies for HPV vaccination schedule completion in immunocompromised girls, including girls living with HIV: Qualitative insights from Eswatini, Malawi, and Uganda' has been provisionally accepted for publication in PLOS Global Public Health.

Best regards,

Edina Amponsah-Dacosta, Ph.D., MPH

Academic Editor